# Designing a Functional CNT+PB@MXene-Coated Separator for High-Capacity and Long-Life Lithium–Sulfur Batteries

**DOI:** 10.3390/membranes12020134

**Published:** 2022-01-23

**Authors:** Guiling Wang, Jiaojiao Li, Zhiling Du, Zhipeng Ma, Guangjie Shao

**Affiliations:** 1College of Chemistry and Materials Engineering, Anhui Science and Technology University, Bengbu 233030, China; wangguilingcg@126.com (G.W.); zhilingdu@hebeu.edu.cn (Z.D.); 2State Key Laboratory of Metastable Materials Science and Technology, Yanshan University, Qinhuangdao 066004, China; lijiao20180207@163.com; 3Hebei Key Laboratory of Applied Chemistry, College of Environmental and Chemical Engineering, Yanshan University, Qinhuangdao 066004, China; 4Hebei Key Laboratory of Air Pollution Cause and Impact (Preparatory), School of Energy and Environmental, Hebei University of Engineering, Handan 056038, China

**Keywords:** lithium–sulfur battery, separator modification, Fe-based Prussian blue, ultra-thin MXene nanosheet

## Abstract

Separators, as indispensable parts of LSBs (lithium–sulfur batteries), play a cucial role in inhibiting dendrite growth and suppressing the shuttle of lithium polysulfide (LiPSs). Herein, we prepared a functional carbon nanotube (CNT) and Fe-based Prussian blue (PB)@MXene/polypropylene (PP) composite separator using a facile vacuum filtration approach. The CNTs and MXene nanosheets are excellent electronic conductors that can enhance the composite separator electrical conductivity, while Fe-based Prussian blue with a rich pore structure can effectively suppress the migration by providing physical space to anchor soluble LiPSs and retain it as cathode active material. Additionally, MXene nanosheets can be well attached to Fe-based Prussian blue by an electrostatic interaction and contribute to the physical barriers that inhibit the shuttle of long-chain soluble Li_2_S_n_ (4 ≤ *n* ≤ 8). When used as a lithium–sulfur (Li–S) cell membrane with a functional coating layer of CNT+PB@MXene facing the cathode side, the batteries reveal a high initial discharge capacity (1042.6 mAh g^−1^ at 0.2 C), outstanding rate capability (90% retention of capacity at 1.0 C) and high reversible capacity (674.1 mAh g^−1^ after 200 cycles at 1.0 V). Of note, separator modification is a feasible method to improve the electrochemical performance of LSBs.

## 1. Introduction

Lithium–sulfur batteries (LSBs) have been regarded as one of the most promising candidates for next-generation rechargeable batteries due to their high theoretical specific capacity of 1675 mAh g^−1^, and outstanding energy density of 2500 Wh kg^−1^, which is up to five times higher compared to lithium-ion batteries [1,2,3]. Additionally, the cathode sulfur processes the favorable characteristics of low cost, great abundance, and excellent environmental compatibility [4,5,6], and so the Li–S cell is more expected to be commercialized. However, its poor cycle life, severe fast capacity fading, and large volume expansion have so far hampered the commercial application. These factors are mainly attributed to the shuttle effect of dissolved lithium polysulfide (LiPS) intermediates (Li_2_S_n_, 4 < *n* ≤ 8) between the anode and the cathode, the insulating nature of sulfur in the cathode, and lithium crystals formed in the anode [7,8,9].

Various strategies have been employed to conquer these difficulties and optimize the electrochemical performance of LSBs [10,11]. Moorthy et al. synthesized a SnS_2_-modified separator for Li–S cells and the battery showed an enhanced areal capacity of 4.03 mAh cm^−2^ by a rapid conversion of LiPSs induced by SnS_2_ [12]. Zhang et al. discovered the Li–S battery with a single-atom Fe-coated separator with a remarkably improved rate capability, and the presence of a Fe single-atom not only increased the binding energy of polysulfide species but also promoted the polysulfide conversion [13]. In recent years, metal-organic frameworks (MOFs), as porous crystalline inorganic–organic materials with high porosity and uniform pore size distribution, have attracted considerable attention in LSBs. Pan et al. prepared mixed matrix membranes based on UiO-66-SO_3_Li MOFs to improve the electrochemical property of LSBs by inhibiting the shuttle of soluble LPS intermediates and lithium dendrite growth since the anionic channels with suitable size of <6 Å from the MOFs can provide the physical barrier, electrostatic repulsion, and intrinsic regular arrangement [14]. Among various MOF porous materials, Fe-based Prussian blue is considered an excellent electron transport medium for which cyanide groups can transfer electrons through different valence state iron ions [15]. Furthermore, it has a unique open frame structure that can provide open channels to facilitate rapid ion conduction, while the poor conductivity greatly restricts its extensive applications [16]. Therefore, it is necessary to introduce materials with good conductivity, and MXene, as a new two-dimensional (2D) material with superior electronic conductivity, has been used widely in LSBs [17]. Recently, Jiang et al. prepared a novel 2D MXene/MOF-derived 2D hybrid (N-Ti_3_C_2_/C), and the exposed MXene surfaces with sufficient Lewis acidity and higher electrical conductivity contributed to the good electrochemical performance in lithium–sulfur batteries [18].

Herein, we report a feasible approach of the vacuum filtration method to design a functional CNT+PB@MXene-Coated/PP composite separator with enhanced electrochemical performance in LSBs. The CNT and MXene ingredients in the separator improve the electronic conductivity and stability due to their intrinsic characteristics of outstanding electrical conductivity and mechanical capacity, while the Fe-based Prussian blue suppresses the migration of dissolved LiPSs by the physical barrier, chemical adsorption and catalytic transformation, which thus contribute to the overall electrochemical performance. Therefore, the prepared CNT+PB@MXene/PP functional porous separator significantly improves the electrochemical performance of lithium–sulfur batteries.

## 2. Experimental Section

Fe-based Prussian Blue Preparation: First, 7.6 g polyvinylpyrrolidone (PVP) and 0.22 g potassium ferrocyanide trihydrate (K_4_Fe(CN)_6_·3H_2_O) were dissolved in 100 mL hydrochloric acid solution (0.1 mol L^−1^). Secondly, the mixed solution was stirred continuously for 30 min to form a homogeneous solution and then transferred to a 100 mL Teflon-lined stainless-steel autoclave for a hydrothermal process at 80 °C for 24 h to obtain Fe-based Prussian blue.

Ultra-thin MXene Nanosheets Preparation: 1.98 g lithium fluoride (LiF) was dissolved in 20 mL hydrochloric acid solution (9 mol L^−1^), and then 2 g MAX powder (Ti_3_AlC_2_) was slowly added to the above solution. Afterwards, the reaction lasted for 24 h at 40 °C under stirring. Upon completion of the reaction, the mixture was rinsed several times with deionized water by centrifugation at 3500 rpm min^−1^, and subsequently dried using the freeze-dried method to get the MXene (Ti_3_C_2_T_x_). Then, 100 mg dried MXene was dissolved in 10 mL distillation, and then centrifuged for 1 h at 3500 rpm min^−1^. After centrifugation, the ultra-thin MXene nanosheet was gained in the form of the supernatant.

Functional Separator Preparation: 1 mg CNT and 1 mg Fe-based Prussian blue (PB) were dissolved in 30 mL anhydrous ethanol solution, respectively. Then, 1 mg ultra-thin MXene nanosheet solution was added into Fe-based Prussian blue solution and then treated by ultrasonication to uniformly disperse. Subsequently, CNT solution, PB and MXene mixed solution were successively pumped onto the polypropylene (PP) membrane to form a uniform and dense CNT+PB@MXene intermediate layer (CNT+PB@MXene) by simple vacuum filtration method. Finally, the prepared CNT+PB@MXene/PP separator was dried in a vacuum at 40 °C for 12 h, and then cut into circular disks of 16 mm.

Composite Sulfur Cathodes Preparation: First, the sublimated sulfur and acetylene black were compounded at 155 °C for 12 h at a mass ratio of 7:3, and then the carbon–sulfur composite and PVDF adhesive were ground to a uniform slurry at a mass ratio of 9:1 by dripping NMP solution. The prepared uniform slurry was coated onto an aluminum foil with the coating thickness of about 200 μm using the doctor blade approach and dried at 60 °C for 12 h in a vacuum oven. After drying, the sulfur cathodes were cut into circular disks of 10 mm with the sulfur loading of 1 mg cm^−2^. Additionally, the cathode with a large sulfur loading of 6.1 mg cm^−2^ was prepared using carbon cloth as a current collector.

Lithium Sulfur Cells Preparation: CR2025-type stainless steel coin cells were thoroughly dried before use and the cells were assembled in an argon-filled glove box. Li_2_S_6_ was prepared as per our previous work and used as electrolyte [19], while lithium metal foil with a diameter of 14 mm was used as anode material and composite sulfur was used as a cathode. The typical assembly process of the battery is as follows: First, the CR2025 negative shell was placed on the template, and then the lithium sheet was placed on the negative shell. Next, the CNT+PB@MXene/PP separator with a diameter of 16 mm was prepared and placed on the lithium sheet, and it was wetted by dripping about 55 μL of the Li_2_S_6_ electrolyte. After that, the gasket, shrapnel and CR2025 positive shell were put in in turn, and finally sealed.

Symmetrical Batteries Preparation: CNT, PB, MXene, PB@MXene and CNT+PB@MXene were mixed with PVDF at a mass ratio of 9:1, respectively, and the other steps were the same as that of the sulfur composite cathode. The electrolyte was 1 mol L^−1^ Li_2_S_6_ with 20 μL DOL/DME (*v/v* = 1:1) solution as additive. Symmetrical batteries were tested by cyclic voltammetry (CV) at a scan rate of 10 mV s^−1^ with a voltage window from −0.8 to 0.8 V.

Structure characterization: The scanning electron microscope (SEM) images were obtained on a SUPRA 55 electron microscope. The powder X-ray diffraction (XRD) patterns were recorded using a RigakuD/max-2500 X-ray diffractometer with Cu Kα radiation (λ = 0.1541 nm, operated at 40 kV and 40 mA). Transmission electron microscopy (TEM) was carried out with the HT 7700 at an acceleration voltage of 100 kV. Termogravimetric analysis (Diamond TG/DTA) was done with the DSC8000 to analyze the content of sulfur in the carbon–sulfur cathode. Ex-situ UV/Vis spectra were obtained with a TU-1810 spectrometer.

Electrochemical measurements: Cyclic voltammetry (CV) was measured in the potential range of 1.7–2.8 V (vs. Li/Li^+^) at a scan rate of 10 mV s^−1^ by a CHI 660E electrochemical workstation (CH Instrument, Chenhua Co., Shanghai, China). Electrochemical impedance spectroscopy (EIS) was carried out in the range of 0.01 Hz–1000 kHz at a potentiostatic signal amplitude of 5 mV on a CHI 660E electrochemical workstation. The galvanostatic charge–discharge curves were recorded within 1.7 and 2.8 V (vs. Li/Li^+^) on a LAND CT2001A battery test system (Wuhan, China).

## 3. Results and Discussion

The XRD pattern of Fe-based Prussian blue in Figure 1 shows obvious diffraction peaks at 17.55°, 24.57°, 35.26° and 39.71°, which corresponded to the (200), (220), (400) and (420) crystal planes of the Prussian blue (PDF # 01-0239), respectively. The results certify that Fe-based Prussian blue MOF was successfully prepared. Figure 2a shows optical images of CNT+PB@MXene/PP (left) and PP separator (right). It can be seen that the CNT+PB@MXene layer is evenly attached to the PP membrane and it presents a uniform and smooth surface. In addition, the CNT+PB@MXene/PP composite membrane remained undestroyed and attached after about 180° bends, noted in Figure 2b, indicating the excellent adhesion and flexibility.

Figure 2c,d are the SEM images of the PP membrane and CNT+PB@MXene/PP membrane, respectively. It can be seen that the PP membrane surface shows lots of pores with an average size of several hundred nanometers. Figure 2d presents a lamellar-coated cube structure, which demonstrates that Fe-based Prussian blue cubes are coated by MXene nanosheets. MXene nanosheets can be well attached to Fe-based Prussian blue by electrostatic interactions as shown in Figure 2e. Fe-based Prussian blue as a MOF material has a rich three-dimensional (3D) pore structure, which offers enough physical space to store dissolved LiPSs and retain them as cathode active material, while MXene nanosheets can act as physical barriers to control the migration of LiPSs [20,21]. In order to further study the CNT+PB@MXene/PP membrane, we also observed the cross-section of the membrane in Figure 2f. The cross-sectional SEM image shows that the thickness of the CNT+PB@MXene intermediate layer is about 20 μm.

Figure 3 shows Li_2_S_6_ adsorption images of the control group (Li_2_S_6_), CNT+PB@MXene, PB@MXene and PB from left to right. The adsorption experiments were carried out as follows: 10 mg PB, PB@MXene and CNT+PB@MXene were added to 5 mL 5 mmol L^−1^ Li_2_S_6_ solution, respectively, and then recorded every hour by taking photos. The results show that the color of Li_2_S_6_ solution with CNT+PB@MXene changed from dark yellow to yellow after 1 h, then to light yellow after 2 h, and then became colorless after 3 h, confirming the excellent Li_2_S_6_ adsorption capacity of CNT+PB@MXene. It took an hour for Fe-based Prussian blue to turn from yellow to colorless. Therefore, Fe-based Prussian blue is a verified effective adsorbent and has strong chemisorption for LiPSs.

In order to further verify the Li_2_S_6_ adsorption capacity of the CNT+PB@MXene layer, the non-in-situ UV/Vis spectra are shown in Appendix A. The spectra manifest that the absorbance of CNT+PB@MXene is significantly lower than that of pure Li_2_S_6_ solution within the characteristic region of Li_2_S_6_, which proves that CNT+PB@MXene has a strong chemical adsorption for Li_2_S_6_ and this is consistent with the results of adsorption experiments in Figure 3.

In order to investigate the effectiveness of the CNT+PB@MXene/PP separator, the corresponding electrochemical characteristics were executed by CR2025-type coin cells in Figure 4. Figure 4a are the CV curves of LSB with the CNT+PB@MXene/PP separator at a scan rate of 0.1 mV s^−1^. The CV curves show two cathodic peaks at 2.02 V and 2.31 V, corresponding to the reduction reaction of cyclo-S8 to long-chain soluble Li_2_S_n_ (4 ≤ *n* ≤ 8) and further to solid Li_2_S_2_/Li_2_S, respectively, while two cathodic peaks near 2.31 V and 2.4 V are attributed to oxidation of Li_2_S_2_/Li_2_S back to S8. The results show that the voltage difference of redox peaks in the CV curve of LSB using the CNT+PB@MXene/PP separator is only 0.29 V, indicating that the electrode has almost no polarization. In addition, no significant current or potential changes occur after 3 cycles, which indicated good electrochemical reversibility [22]. 

Figure 4b shows the charge–discharge curves of LSBs with CNT+PB@MXene/PP and PP separators at different rates. When CNT+PB@MXene/PP was used as a separator, the discharge curves showed two typical discharge platforms, which agreed well with the CV results. It is worth noting that discharge capacity (Q_H_) from the higher discharge platform at 2.33 V at 0.2 C is used to evaluate the generation degree and the capture ability of polysulfide in LSBs, while the lower discharge platform is the main discharge capacity [23]. The smooth and highly overlapping discharge platform from 0.2–2.0 C confirmed the superior reversibility of the solid redox reaction. The distinguished electrochemical properties are ascribed to the rich pore structure and strong capability to capture LiPSs of Fe-based Prussian blue, as well as the excellent electrical conductivity of cross-stacked carbon nanotubes and ultra-thin MXene nanosheets. As shown in Figure 4c, the LSB with the CNT+PB@MXene/PP separator displays outstanding discharge capacities of 1042.6, 868.5, 732.7 and 572.6 mAh g^−1^ at 0.2, 0.5, 1.0 and 2.0 C, which are higher than the values of 652.3, 550.8, 539.6 and 476.3 mAh g^−1^ for the PP separator at the same current densities. Additionally, it reveals an excellent rate performance with near 90% retention of capacities from 732.7 and 572.6 mAh g^−1^ to 640.8 and 555.9 mAh g^−1^ after current density returned to 1.0 and 2.0 C, while the specific capacity of LSB with a PP separator can just regain 523.6 and 281.1 mAh g^−1^. It can be inferred that CNTs with high electronic conductivity, ultra-thin MXene nanosheets with high specific surface area and Fe-based Prussian blue with rich porous structure can effectively inhibit the migration of polysulfide and accelerate the conversion of long-chain soluble Li_2_S_n_ (4 ≤ *n* ≤ 8) and further to solid Li_2_S_2_/Li_2_S.

Electrochemical impedance spectroscopy (EIS) was executed to explore the kinetics of LSBs with CNT+PB@MXene/PP as a separator at the electrode/electrolyte interface shown. The Nyquist plots consisted of a semicircle at the high frequency and a sloping line at the low frequency, which corresponded to the charge transfer resistance (Rct) and Warburg impedance (Zw), respectively. It is noted that the plot of LSB with the CNT+PB@MXene/PP separator presents a significantly lower semicircle than that of pure PP separator. The results suggest that the battery with the CNT+PB@MXene/PP separator has a small charge transfer resistance of about 25.45 Ω, which is due to the high conductivity of carbon nanotubes and ultra-thin MXene nanosheets. Long-term stability of LSB with the CNT+PB@MXene/PP separator was tested by galvanostatic charge–discharge at 1.0 C (Figure 4e). The CNT+PB@MXene/PP battery displayed a discharge capacity of 941.7 mAh g^−1^ and it still maintained a high reversible capacity of 674.1 mAh g^−1^ after 200 cycles at 1.0 C, where the capacity retention rate went up to 71.6%. In contrast, the discharge capacity of PP started at 440.6 mAh g^−1^ and displayed a gradual increase for the first 20 cycles, which resulted from an electrochemical activation process [24,25]. This might have been caused by the good wettability of PB CNT and MXene [26,27], so the cycling curve of the CNT+PB@MXene-modified separator did not appear to show an increase in the discharge capacity. However, the poor wettability of PP results in a certain amount of time required for electrode activation in the initial charge–discharge process. The high specific capacity and good cycle stability of the battery are ascribed to the rich pore structure of Fe-based Prussian blue, which can effectively adsorb LiPSs and facilitate its conversion to lithium sulfide. Moreover, high conductivity carbon nanotubes and ultra-thin MXene nanosheets further accelerate the catalytic conversion kinetics of polysulfide. A Li–S half-cell with a high sulfur loading of 6.1 mg cm^−2^ was assembled to evaluate the practical application of the CNT+PB@MXene/PP separator in Li–S cells. Appendix A shows the cycling stability of the Li–S half-cell with the CNT+PB@MXene/PP separator performed on galvanostatic charge–discharge at 0.1 C. The half-cell revealed a high initial discharge specific capacity of 851.5 mAh g^−1^ (5.2 mAh cm^−2^), with about 95.1% capacity retention rate after 60 cycles. The result demonstrates that Li–S cells with high sulfur loading can reveal outstanding excellent electrochemical performance when CNT+PB@MXene/PP was used as a separator.

In order to verify the conversion kinetics of long-chain soluble Li_2_S_n_ (4 ≤ *n* ≤ 8) to solid Li_2_S_2_/Li_2_S, cyclic voltammetry (CV) tests were carried out on symmetric cells with different electrode materials of CNT, PB, MXene, PB@MXene and CNT+PB@MXene in 0.5 mol L^−1^ Li_2_S_6_ electrolyte. The CV curves in Figure 5 show that CNT and MXene electrodes present a small current response, while the PB electrode showed no current response, indicating that CNT and MXene possess catalytic conversion capacity of LiPSs. The CV curve of the CNT+PB@MXene electrode in Figure 5 shows a higher current density than that of CNT, PB and MXene electrodes at the same potential, meaning more activity derived from the CNT+PB@MXene electrode. In addition, a pair of reversible redox peaks at 0.35 and −0.35 V appeared on its curve, verifying that the CNT+PB@MXene layer has strong catalytic conversion ability for polysulfide to lithium sulfide.

It is well-known that the shuttle effect of dissolved polysulfide can result in the self-discharging behavior of batteries. Figure 6a,b show the self-discharge curves of the PP membrane and CNT+PB@MXene/PP membrane at 2.8 V. As can be seen in Figure 6a,b, the capacity decay of the Li–S cell with PP membrane is 467 mAh g^−1^, while that of the Li–S cell with CNT+PB@MXene/PP membrane is only 192.7 mAh g^−1^ after a 14-day rest. To further identify the mechanism of functional CNT+PB@MXene layer, the cells were rested at a voltage of 2.06 V as shown in Figure 6c,d since soluble polysulfide content reached the highest at the beginning of the second discharge platform. The charge–discharge curves show that the capacity decay with the CNT+PB@MXene/PP membrane in Figure 6d reduces to 153.1 mAh g^−1^ after a 10-day rest, which is much less than 292.9 mAh g^−1^ of the Li–S cell with the PP membrane in Figure 6c. The result indicates that the CNT+PB@MXene layer on the membrane in LSBs can alleviate self-discharge behavior by reducing the shuttle effect by the rich pore structure of Fe-based Prussian blue and physical barriers of MXene nanosheets.

## 4. Conclusions

In summary, a functional composite separator was fabricated by introducing a CNT+PB@MXene coating layer on the PP membrane. Functional modification of a commercial PP membrane allows the integration of cross-stacked carbon nanotubes, Fe-based Prussian blue and ultra-thin MXene nanosheets. The rich pore structure and strong capability of Fe-based Prussian blue significantly suppresses migration of polysulfide and accelerates its conversion to solid lithium sulfide. Furthermore, high conductivity carbon nanotubes and ultra-thin MXene nanosheets effectively accelerate the catalytic conversion kinetics of polysulfide. Therefore, this work demonstrates the importance of rational design of the functional separator for achieving high-performance LSBs.

## Figures and Tables

**Figure 1 membranes-12-00134-f001:**
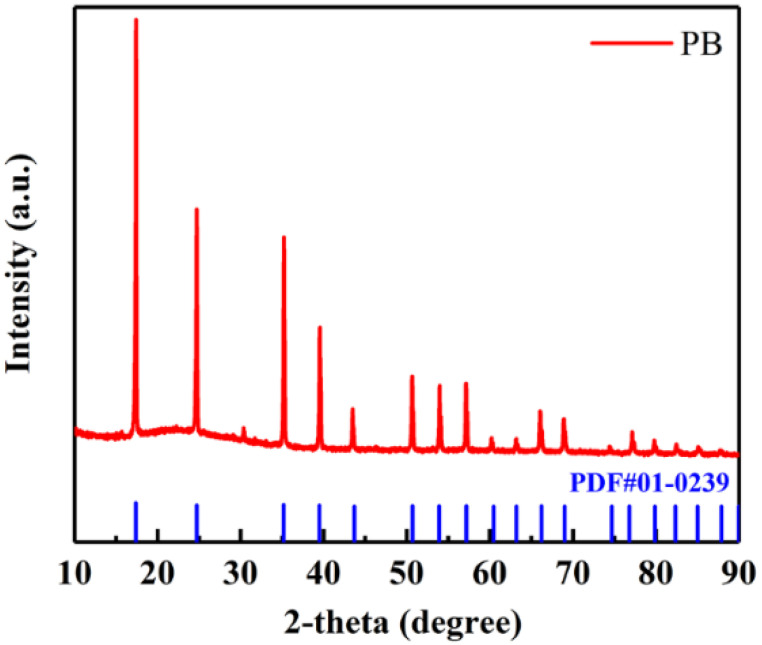
XRD pattern of Fe-based Prussian blue MOF.

**Figure 2 membranes-12-00134-f002:**
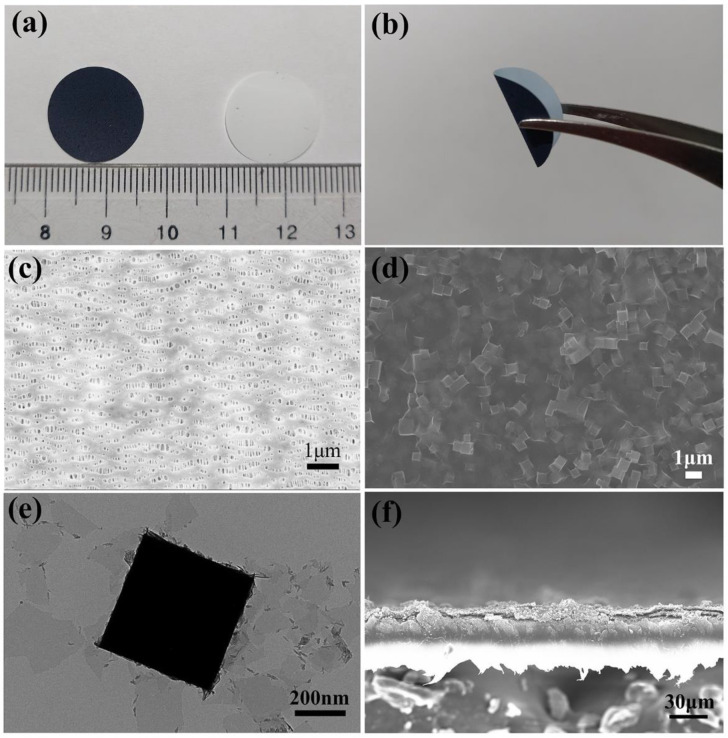
(**a**) Optical images of CNT+PB@MXene/PP (left) and PP separator (right), (**b**) Bend image of CNT+PB@MXene/PP, (**c**,**d**) SEM images of PP, CNT+PB@MXene/PP, (**e**) TEM of MXene@PP, (**f**) Cross-sectional SEM image of CNT+PB@MXene/PP.

**Figure 3 membranes-12-00134-f003:**
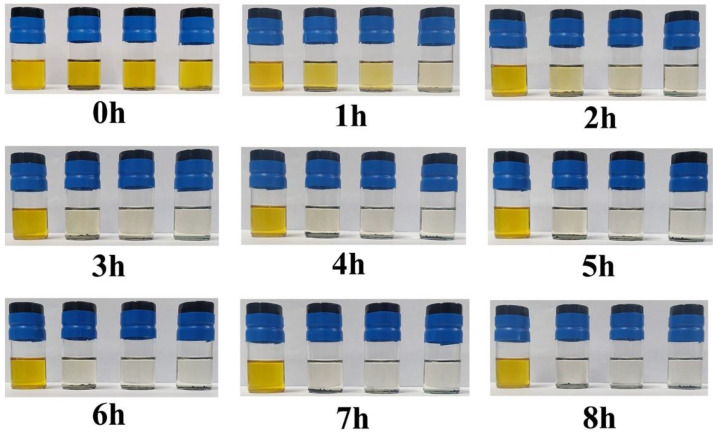
Time-dependent Li_2_S_6_ adsorption images of control group (Li_2_S_6_), CNT+PB@MXene, PB@MXene and PB from left to right.

**Figure 4 membranes-12-00134-f004:**
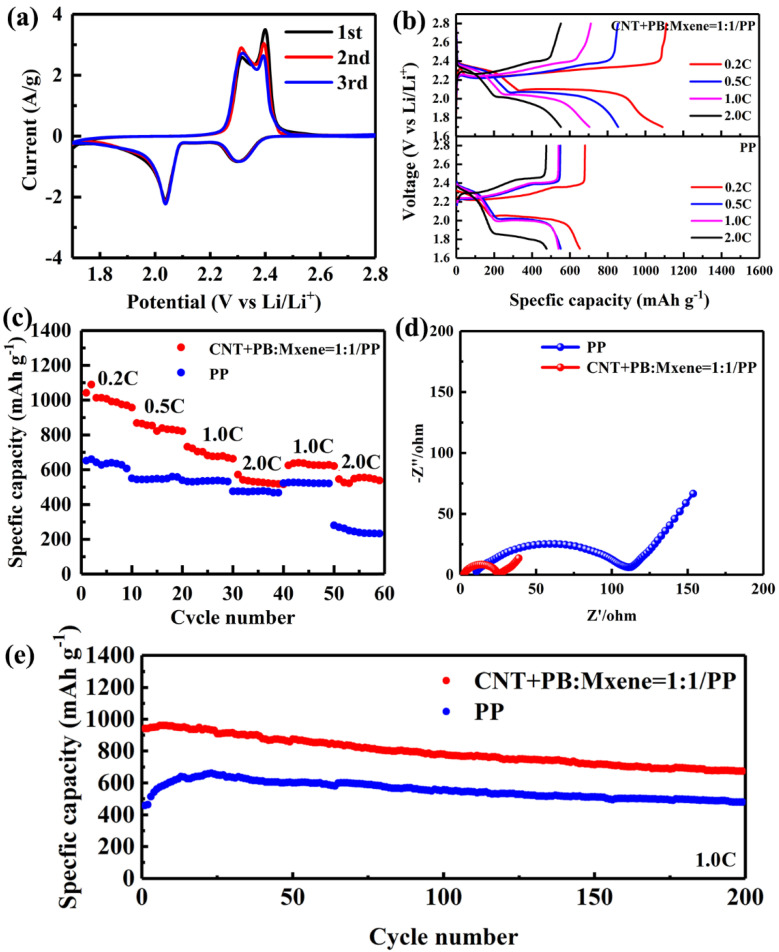
CV curves (**a**) at a scan rate of 0.1 mV s^−1^ of LSB with the CNT+PB@MXene separator. Galvanostatic charge-discharge profiles (**b**) and rate performance (**c**) with CNT+PB@MXene/PP (above) and PP (below) separators at various C-rates from 0.2 C to 2.0 C. EIS profiles (**d**) and long-term cycling curves (**e**) with CNT+PB@MXene/PP and PP separators.

**Figure 5 membranes-12-00134-f005:**
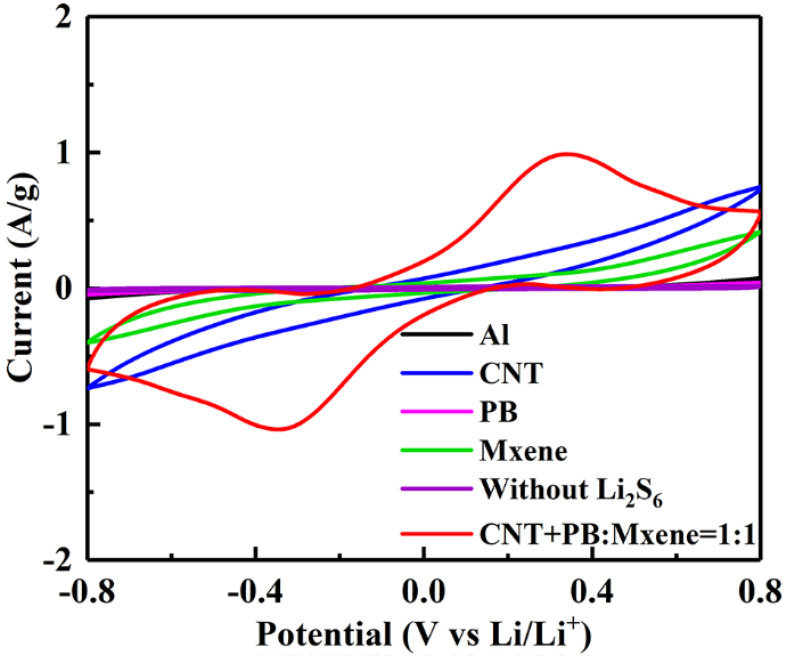
CV curves of symmetric LSBs at a scan rate of 10 mV s^−1^ with CNT, PB, MXene, PB@MXene and CNT+PB@MXene as electrode materials.

**Figure 6 membranes-12-00134-f006:**
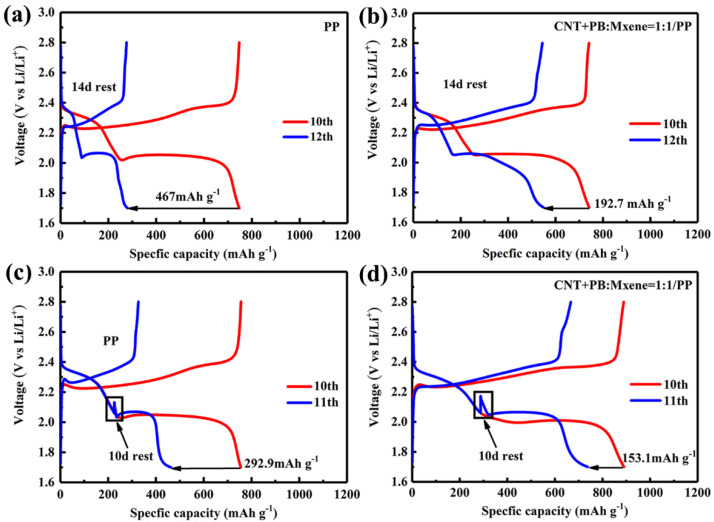
Self-discharging curves of LSBs with different separators of PP (**a**,**c**) and CNT+PB@MXene (**b**,**d**) after 14- and 10-day rest at 2.8 and 2.06 V, respectively.

## Data Availability

Not applicable.

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
