# Peer review of "Designing a Functional CNT+PB@MXene-Coated Separator for High-Capacity and Long-Life Lithium–Sulfur Batteries"

_membranes, 2022, doi:10.3390/membranes12020134_

Round 1
Reviewer 1 Report
Work presented in the paper is interesting and the paper is rather well written. This paper can be recommended for publication after clarifying the queries listed below.
- Give a comparative analysis of the action of different types of smart separator materials in improving the electro-chemical performance of Li-S cells and highlight the significance of the separator material used in the present work
- Different types of interlayers have been used to inhibit polysulphide shuttling effect and on comparing with the action of interlayers in improving the performance of Li-S cells, where does the separator material of the present work stand?
- Have the authors assessed the performance of these Li-S cells at higher C rates like 2C, 5Cand 10C?
Reviewer 2 Report
The research on Li-S batteries is gaining wide attention and this is a timely article. Utilizing Mxenes for Li-S batteries is really interesting as Mxenes have great scope in future due to their physical/chemical properties. I recommend the article for publication after addressing the following changes.
- The electrochemical performance of bare CNT and bare Mxenes coated separators can be studied and compared in Fig 4.
- Why is discharge capacity increases (fig. 4e) when using bare separator ?
- High mass loading testing or testing with least electrolyte volume can studied to increase the novelty of the work.
- The aerial capacity (based on S mass loading) should be calculated and reported.
- Is there any chemical/physical changes in PB/Mxene coated separator after cycling?
- What would happen if PB/Mxene mixture is used as host for sulfur cathode? There are lot of recent reports on using Mxenes as Sulfur host. But I haven’t found something with PB as Sulfur cathode host.
- The electrolyte – separator interaction can be studied with contact – angle measurements.
- Following related reference on Li-S work can be added : (i) Nanoscale horizons 4 (1), 214-222; (ii) Nanomaterials 10 (6), 1220; (iii) Sustainable Energy Fuels, 2020,4, 3500-3510; (iv) ACS Appl. Mater. Interfaces 2019, 11, 28, 25147–25154
Reviewer 3 Report
This is an interesting publication and there is no doubt that the proposed solution helps to improve performance of LSB to a certain extent. However, modification of the cathode-separator interface for entrapment and conversion of soluble polysulfides is not an ultimate solution, as the conversion should take place within the cathode and not at the interface, to avoid degradation of the cathode.
p.5 l.172 Instead of “Non-in-situ” use “ex situ.”
